# Skipping Breakfast and a Meal at School: Its Correlates in Adiposity Context. Report from the ABC of Healthy Eating Study of Polish Teenagers

**DOI:** 10.3390/nu11071563

**Published:** 2019-07-11

**Authors:** Lidia Wadolowska, Jadwiga Hamulka, Joanna Kowalkowska, Natalia Ulewicz, Magdalena Gornicka, Marta Jeruszka-Bielak, Małgorzata Kostecka, Agata Wawrzyniak

**Affiliations:** 1Department of Human Nutrition, Faculty of Food Sciences, University of Warmia and Mazury in Olsztyn, Sloneczna 45F, 10-718 Olsztyn, Poland; 2Department of Human Nutrition, Faculty of Human Nutrition and Consumer Sciences, Warsaw University of Life Science—SGGW, Nowoursynowska 159 C, 02-776 Warsaw, Poland; 3Department of Chemistry, Faculty of Food Science and Biotechnology, University of Life Sciences, 15 Akademicka Street, 20-950 Lublin, Poland

**Keywords:** meals skipping, breakfast, meal at school, central adiposity, obesity, diet quality, lifestyle

## Abstract

Little is known on skipping breakfast and a meal at school, especially considered together. The study identified nutrition knowledge-related, lifestyle (including diet quality, physical activity, and screen time) and socioeconomic correlates of skipping breakfast and a meal at school, considered together or alone and assessed the association of skipping these meals with adiposity markers in Polish teenagers. The sample consisted of 1566 fourth and fifth grade elementary school students (11–13 years). The study was designed as a cross-sectional study. Data related to the consumption of selected food items and meals, physical activity, screen time, sociodemographic factors, and nutrition knowledge (all self-reported) were collected (in 2015–2016) with a short form of a food frequency questionnaire. Respondents reported the usual consumption of breakfast (number of days/week) and a meal or any food eaten at school (number of school days/week) labelled as ‘a meal at school’. The measurements of body weight, height, and waist circumference were taken. BMI-for-age ≥25 kg/m^2^ was considered as a marker of overweight/obesity (general adiposity), while waist-to-height ratio ≥0.5 as a marker of central obesity (central adiposity). A multivariate logistic regression was applied to verify the association between variables. A total of 17.4% of teenagers frequently skipped breakfast (4–7 days/week), 12.9% frequently skipped a meal at school (3–5 school days/week), while 43.6% skipped both of these meals a few times a week. Predictors of skipping breakfast and/or a meal at school were female gender, age over 12 years, urban residence, lower family affluence, lower nutrition knowledge, higher screen time, and lower physical activity. In comparison to “never-skippers,” “frequent breakfast skippers” were more likely to be overweight/obese (odds ratio, OR 1.89; 95% confidence interval, 95%CI 1.38, 2.58) and centrally obese (OR 1.63; 95%CI 1.09, 2.44), while skippers a few times a week of both of these meals were more likely to be overweight/obese (OR 1.37; 95%CI 1.06, 1.78). Concluding, we estimated that a large percentage of Polish teenagers (approx. 44%) usually skipped both breakfast and a meal at school a few times a week. Similar predictors of skipping breakfast and predictors of skipping a meal at school were identified. Special attention should be paid to promoting shortening screen time and increasing physical activity and teenagers’ nutrition knowledge which are relatively easily modifiable correlates. The study shows that skipping both of these meals a few times a week was associated with general adiposity and also strengthens previous evidence showing the association of frequent skipping breakfast with general and central adiposity.

## 1. Introduction

Regular consumption of meals, starting the day with breakfast, is a basic dietary recommendation especially for children and adolescents. Many adverse effects resulting from skipping breakfast have been described. In students aged 6–20 from Europe (France, Spain, Ireland), United States, and Japan, breakfast skipping was associated with poorer diet quality [1,2,3,4], worse mood [5,6], educational progress, and cognitive function development, including worse academic achievements [7,8,9] and increased risk of obesity [10,11,12,13,14,15,16], markers of insulin resistance [10,17], and cardiometabolic risk factors [10,11,15,18]. The poorer diet quality of breakfast skippers was attributed to worse food choices, e.g., lower consumption of dairy foods, cereal products, fruits and vegetables, and higher consumption of energy-dense snack foods, which resulted in lower intakes of vitamins, minerals, fiber, and protein [2,7]. There is inconsistency in regard to the diet energy value and there is no clear explanation of the mechanism promoting excessive adiposity in breakfast skippers [10,11,19]. The negative role of lower physical activity in energy balance among breakfast skippers was discussed [20,21,22] and further research into various forms of active and inactive behaviors of breakfast skippers is needed. 

Less research was focused on the second eating episode of the day eaten by school-age children. During school days, this meal is usually eaten at midday, as lunch or ‘second breakfast’ (more typical in Poland), prepared at home and brought to the school or prepared in the school canteen (free of charge or not), or bought in a school shop, e.g., snacks or sandwiches. To date, most of the research is focused on outcomes of school-meal programs, often provided free of charge to students from low-income families [23,24,25]. In Poland, some national programs supporting health-promoting dietary habits of school-age children are on-going (e.g., ‘Fruits in School’ or ‘A Glass of Milk’) [26], and government programs addressed to students from low-income families have also been implemented (e.g., ‘A meal at school and at home’ in 2019–2023) [27] but no scientific evaluation of the programs’ outcomes was completed. Moreover, the association of a meal consumed at school with lifestyle and body weight was not clearly established. Students who skipped a meal at school, including double breakfast eaters, were less likely to have a normal weight than regular consumers [28]. However, a positive, but insignificant, association between body mass index (BMI) and energy intake from school meals was reported [29]. Two possible explanations were discussed in respect to these findings. Students skipping a meal at school could be less physically active and expend less energy throughout the day than school meal consumers [28,29]. Secondly, more caloric meals consumed in the morning could be compensated by less caloric meals later in the day, making it easier to maintain energy balance by regular meal consumers [30]. 

From an international perspective, skipping breakfast was reported in 16%–23% of 9–11-year-old children from 12 countries across the world [16]. However, other data were also reported due to the significant variation in the definition of breakfast and breakfast skippers [2,3,4,10]. In the Health Behavior in School-aged Children (HBSC) study, a significant decrease in daily breakfast consumption over eight years (from 2002 to 2010) was found in adolescents from 11 European countries, including Poland [31]. In Polish adolescents, irregular or very rare breakfast consumption was found in 28%–44% of those aged 11–13 years and 46%–52% of those aged 13–18 years [32,33]. Skipping breakfast across the world was more frequently reported in girls, older school-age children, those with lower family social economic status, low family functioning, and families with a single parent [7,10,11,14,30,31,34,35]. Numerous reasons for skipping breakfast were given, including the lack of time in the morning, no desire to eat food, or body weight control, especially by females [5,11,36]. Data regarding the prevalence of skipping a meal eaten at school is limited, and data regarding both breakfast and a meal eaten at school is also missing. To date, predictors of skipping both of these meals considered together have not been explored.

The aims of the study were (1) to identify nutrition knowledge-related, lifestyle (including diet quality, physical activity, and screen time) and socioeconomic correlates of skipping breakfast and a meal at school, considered together or alone, (2) to assess the association of skipping these meals with adiposity markers in Polish teenagers. It was decided to focus on 11–13-year-old students because the age of 10(12) years old is considered a breakthrough [37]. At this age, changes start in the perception of oneself as an independent individual and food consumer. Thus, early adolescence may be the last moment for implementing nutrition education before determining young people’s relatively stable eating habits.

## 2. Materials and Methods

### 2.1. Ethical Approval

The project followed the ethical standards recognized by the Helsinki Declaration. The study protocol was registered and approved by the Bioethics Committee of the Faculty of Medical Sciences, University of Warmia and Mazury in Olsztyn (Poland) on 17 June 2010, Resolution No. 20/2010). Parental or legal guardians’ written informed consent was obtained for the participation of their children.

### 2.2. Study Design

The study was designed as a cross-sectional study. Data related to the study were collected (in 2015–2016) by researchers from eight academic centers, covering the entire territory of Poland, as part of the national multicenter ‘ABC of Healthy Eating’ project. More details on the study protocol and methods were described previously [38]. The study was explained to the participants before the start, i.e., to the pupils at the school lessons and to the parents or legal guardians during the meetings. All data were collected by experienced researchers (well-trained in taking anthropometric measurements and collecting dietary data) at schools instead of regular school lessons. Students who refused to participate attended other school activities. The teachers were present during the research but did not take part in data collection.

Anthropometric measurements were taken. Data related to meal consumption, lifestyle (including diet quality, physical activity and screen time), nutrition knowledge, and socioeconomic variables (all self-reported) were collected with a questionnaire (acronym: SF-FFQ4PolishChildren) dedicated to school-age children. The questionnaire was developed by Kowalkowska, Wadolowska, and Hamulka for the ‘ABC of Healthy Eating’ project. The internal compatibility of the questionnaire was tested for 630 teenagers aged 11–15 showing acceptable to good reproducibility of this FFQ (data not published, paper under review). The questionnaire was self-administered by teenagers in the classroom and supervised by researchers. Explanations were given if necessary. The completion of the questionnaire took the teenagers approx. 40 min. 

### 2.3. Participants

A convenient sample selection was applied. A school class was the smallest unit in the sample selection. It was decided to start recruitment based on school classes because students were subject to the same school education and would be at a similar stage of development. First, schools were invited to take part in the study and each student of 4th- and 5th-grade classes of this school (classes that met criteria, and teachers gave their approval) were then invited. Recruitment was conducted in elementary schools from urban, sub-urban, and rural areas covering the entire territory of Poland. Details of sample selection were described previously [38]. 

School inclusion criteria were: a location at a convenient distance from the academic centers (up to 50 km),the agreement of the school principal to school participation,no previous participation of the school in other nutrition-health education programs.

Participant inclusion criteria were:written consent of parents or legal guardians to participate,4th- and 5th-grade classes of elementary school; the expected age of the students was 11–12 years in 2015 and 11–13 years in 2016 (due to changes in the Polish law concerning the age of starting education by children);no disability self-declared by parent or legal guardian or teacher.

When cleaning the data, participant exclusion criteria was age <11 or >12 years in 2015 and <11 or >13 years in 2016.

In total, 116 classes from 35 schools were selected across Poland (Figure 1). Initially, 1678 students were recruited. A total of 112 participants were excluded from analyses, 109 participants because of age and three participants due to missing data on consumption of breakfast or a meal at school. The study included 1566 teenagers aged 11–13 years, 759 boys (48.5%) and 807 girls (51.5%).

### 2.4. Breakfast and a Meal at School

Frequency of meal consumption covered the period of 12 last months and reflected long-term habits. Respondents were asked to report their usual frequency of breakfast consumption in a number of days/week and a meal at school in number of school days/week. For breakfast consumption, respondents could choose one of four categories: 0/week, 1–3/week, 4–6/week, 7/week. After distribution analysis, three categories of breakfast skipping were determined (labelled as): 4–7/week (frequent skipping), 1–3/week (skipping a few times a week), and 0/week (never skipping, i.e., every day consumption). For a meal at school consumption, respondents could choose one of four categories: 0/week, 1–2/week, 3–4/week, 5/week. After distribution analysis, three categories of skipping a meal at school were determined (labelled as): 3–5/week (frequent skipping), 1–2/week (skipping a few times a week), and 0/week (never skipping, i.e., every school day consumption). Table 1 presents categorization of skipping breakfast and a meal at school when considered together.

Detailed information regarding breakfast and a meal at school were guidelines to collect data and were both (i) described in short manual for researchers, (ii) given to the participants (verbally) before questionnaire administration. Breakfast was considered as consuming solid food with or without beverages at the first eating episode per day, before or at the start of daily activities [5,10,19,38]. Any meal consumed at school was labelled as ‘a meal at school’ and considered as consuming solid food with or without beverages at the second eating episode of the day during school days, regardless of whether snacks or other foods were eaten. In respect to a meal at school, it did not matter whether the meal was prepared at home and brought to the school or prepared in the school canteen (free of charge or not), or bought in a school shop. Drinking only beverages (e.g., water, tea, fruit juices, or sweetened beverages) was not considered as a meal, except for milk drinks (e.g., milk or yoghurt), if at least one serving was consumed. In general, the amount of food was not taken into consideration but the consumption of very small amount of food (e.g., one bite of bread or a few sips of milk) was not classified as a meal due to its low energy value. 

### 2.5. Diet Quality

Respondents were asked to specify their usual consumption frequency within the 12 last months of eight food items: dairy products (e.g., milk, yoghurt, cottage cheese, cheese), fish (e.g., baked, fried, in vinegar, canned), vegetables (e.g., fresh, boiled, baked, stewed), fruit (fresh or frozen), fast foods (e.g., chips, pizza, hamburgers), sweetened drinks (e.g., cola type, water with syrup, tea type with sugar), energy drinks (the names of many such drinks were given) and sweets or confectionery (e.g., cookies, sweets, cake, bars of chocolate, chocolate). For food frequency consumption respondents could choose one of seven categories (converted into daily frequency, times/day): never or almost never (0), rarely than once a week (0.06), once a week (0.14), 2–4 times/week (0.43), 5–6 times/week (0.79), every day (1), and several times a day (2). 

To assess diet quality with regard to foods, especially of those participants who were engaged in a variety of opposite behaviors, two diet quality scores were established (a priori approach) based on previous knowledge and similar studies [39,40]. Each diet quality score consisted of four food items and was calculated by summing up the daily frequencies of selected food items. The pro-Healthy Diet Index (pHDI) included dairy products, fish, vegetables and fruit, the non-Healthy Diet Index (nHDI) included fast foods, sweetened carbonated drinks, energy drinks, and sweets or confectionery. Each diet quality score (ranged from 0 to 8 times/day) was recalculated to express it in % points (range: 0–100). Three levels of each diet quality score were established: low (<33.33% points), moderate (33.33 to <66.66% points) and high (≥66.66% points). A higher % of points reflected greater adherence to the diet quality score, i.e., for pHDI—better diet quality, and for nHDI—worse diet quality.

### 2.6. Physical Activity and Screen Time

Physical activity was assessed using two questions regarding physical activity at school and at leisure time. The respondents could choose one out of three answers describing their physical activity at school (low, moderate, vigorous) and at leisure time (low, moderate, vigorous). Many examples for each answer were given. Details are presented in Table 2. After combining some categories of both questions, the respondents were divided into three categories: low physical activity, moderate physical activity, and high physical activity. 

A time spent in front of a screen was labelled as “screen time”. Screen time was assessed using the question ‘How much time do you spend watching TV or in front of a computer or tablet on an average day of the week?’ It was explained to participants that time spent sedentary should be considered, thus, time spent with smartphones (e.g., while walking) was not considered. Respondents reported screen time, choosing one of six categories (hours/day): <2, 2 to <4, 4 to <6, 6 to <8, 8 to <10, and ≥10. After combining some answers, three categories of screen time were considered: <2 h/day, 2 to <4 h/day, ≥4 h/day.

### 2.7. Nutrition Knowledge

Nutrition knowledge level was determined based on 18 questions (Appendix A). Participants were asked about nutrition based on questions developed by Whati et al. [41] and adapted to Polish conditions and education [38]. A correct answer was scored with 1 point, wrong or “I don’t know” answers or missing data were scored with 0 points. Points were summed up for each respondent to calculate the nutrition knowledge score. Based on tertiles distribution of nutrition knowledge score, the respondents were divided into three categories: the lowest nutrition knowledge (0–4 points), moderately-low nutrition knowledge (5–7 points), and higher nutrition knowledge (8–18 points).

### 2.8. Socioeconomic Data

The socioeconomic predictors of skipping meals included gender, age, residence, and family affluence. Respondents were divided into two categories of residence (rural or urban). Family affluence was determined by using the Family Affluence Scale (FAS) on the base of the household characteristics—details are presented in Appendix A and were described previously [38]. The scale was developed for the Health Behavior of School-aged Children (HBSC) cross-national study including 44 countries. We used a scale composed of four questions described by the Polish team of the HBSC study [42]. The points were summed up for each respondent (range 0 to 7). Based on FAS distribution, the respondents were divided into three categories: low FAS (0–4 points; <25th quartiles), moderate FAS (5–6 points), high FAS (7 points; ≥75th quartiles).

### 2.9. Adiposity Markers

The measurements of body weight (kg, recoded with a precision of 0.1), height (cm, precision: 0.1 cm) and waist circumference (WC; cm, precision: 0.1 cm) were taken (between 8 and 12 a.m.). Similar professional measuring devices and tapes were used in all research centers. All measurements were taken in light clothing and without shoes according to the ISAK International Standards for Anthropometric Assessment guidelines [43]. Body mass index (BMI, kg/m^2^) and waist-to-height ratio (WHtR) were calculated. BMI-for-age was categorized according to international sex-specific cut-offs for teenagers [44]. BMI-for-age ≥25 kg/m^2^ was used as an overweight/obesity measure (general adiposity marker), and WHtR ≥0.5 as a central obesity measure (central adiposity marker) [18,38].

### 2.10. Statistical Analysis

Sample size calculation was based on the expected ratio of respondents consuming and not consuming breakfast every day (ratio 75/25) and the difference in overweight/obesity prevalence between respondents consuming and not consuming breakfast every day (20% vs. 30%, respectively). With a 5% significance level and 80% power, the minimum sample size required is approximately 904 respondents (i.e., 678/226 respondents consuming/not consuming breakfast every day and an equal number of respondents in each research center) to detect a difference between groups in overweight/obesity prevalence, including recoding error or 10% missing data. Thus, the study sample (*n* = 1566) exceeded the minimum sample size required (*n* = 904).

We analyzed two variable sets: (1) as outcomes (dependent variables) skipping breakfast and a meal at school (together or alone) were considered, while as predictors (independent variables) nutrition knowledge-related, lifestyle (including diet quality, physical activity, and screen time) and socioeconomic correlates were considered; (2) as outcomes (dependent variables) adiposity markers were considered while as predictors (independent variables) skipping breakfast and a meal at school (together or alone) were considered.

Categorical variables were presented as a sample percentage (%). The differences between groups were verified by a chi-square test. The associations between skipping meals and variables under study were verified with a multivariate logistic regression. The odds ratios (ORs) and 95% confidence intervals (95%CI) were calculated. Three analysis were carried out—separately for breakfast or a meal at school and also for both of these meals considered together. The reference category (OR = 1.00) was daily breakfast consumption and/or daily consumption of a meal at school (i.e., never skipping). The ORs were adjusted for gender, age (years), residence (two categories), FAS (points), nutrition knowledge (points), physical activity (three categories), screen time (three categories), excluding the modelled variable from confounders’ set, respectively. The models were also adjusted for the consumption of breakfast or a meal at school as confounders (meal-adjusted models) and are presented in the supporting information. The significance of ORs was assessed by Wald’s statistics. For all tests, the two-sided significance level *p* < 0.05 was considered as significant. Analyses were performed using Statistica software (version 12.0 PL; StatSoft Inc., Tulsa, OK, USA; StatSoft, Krakow, Poland). 

## 3. Results

### 3.1. Breakfast

A total of 17.4% of teenagers frequently skipped breakfast while 12.6% skipped breakfast a few times a week (Table 3 and Appendix A). 

In comparison to never skipping breakfast, frequently skipping breakfast was more likely in teenagers with a screen time ≥2 h/day (OR 1.78; 95%CI 1.29, 2.46 or 2.82; 1.96, 4.06), in those with low FAS (OR 2.45; 95%CI 1.64, 3.66) or moderate FAS (OR 1.54; 95%CI 1.06, 2.23) than high FAS, in 13-year-olds (OR 2.04; 95%CI 1.17, 3.56) than 11-year-olds, in teenagers with the lowest nutrition knowledge score (OR 1.51; 95%CI 1.05, 2.17) than higher nutrition knowledge score, and in girls (OR 1.44; 95%CI 1.09, 1.91) than boys (Table 4). Skipping breakfast a few times a week was less likely in teenagers with moderate/high pHDI (OR 0.57; 95%CI 0.39, 0.83) than those with low pHDI, and was also less likely in teenagers with moderate/high nHDI (OR 0.38; 95%CI 0.16, 0.90) than those with low nHDI. Similar associations were found in meal-adjusted models after including a meal at school as a confounder (Appendix A).

Frequent breakfast skippers were more likely to be overweight/obese (OR 1.89; 95%CI 1.38, 2.58) and centrally obese (OR 1.63; 95%CI 1.09, 2.44) while less likely to fall in the thinness category (OR 0.52; 95%CI 0.29, 0.94), in comparison to never-skippers (Table 5). Similar associations were found in meal-adjusted models after including a meal at school as a confounder (Appendix A).

### 3.2. A Meal at School

A total of 12.9% of teenagers frequently skipped a meal at school while 18.1% skipped this meal a few times a week (Table 3 and Appendix A). 

In comparison to never skipping a meal at school, frequently skipping a meal at school was more likely in teenagers with a screen time ≥4 h/day (OR 2.59; 95%CI 1.77, 3.81) than <2 h/day, in 13-year-olds (OR 1.95; 95%CI 1.06, 3.59) than 11-year-olds, in teenagers with the lowest nutrition knowledge score (OR 2.17; 95%CI 1.39, 3.40) than higher nutrition knowledge score, in teenagers with low physical activity level (OR 1.96; 95%CI 1.11, 3.49) or moderate physical activity level (OR 1.53; 95%CI 1.05, 2.24) than high physical activity level and in urban residents (OR 1.44; 95%CI 1.03, 2.01) than rural residents while less likely in teenagers with moderate/high pHDI (OR 0.69; 95%CI 0.47, 1.00) than those with low pHDI (Table 4). Skipping a meal at school a few times a week, in comparison to never skipping, was significantly associated with female gender, low or moderate physical activity level, and a screen time ≥2 h/day. Similar associations were found in meal-adjusted models after including breakfast as a confounder (Appendix A).

In the logistic regression analysis, no significant association was found for skipping a meal at school and overweight/obesity or central obesity as well as thinness (Table 5 and Appendix A).

### 3.3. Breakfast and a Meal at School

A total of 4.5% frequently skipped both breakfast and a meal at school while 43.6% skipped both of these meals a few times a week (Table 3 and Appendix A).

In comparison to never skipping both of these meals, frequent skipping of both of these meals was more likely in 13-year-old teenagers (OR 5.13; 95%CI 1.88, 14.01) than 11-year-olds, in teenagers with a screen time ≥4 h/day (OR 3.93; 95%CI 2.10, 7.37) than <2 h/day, with low physical activity level (OR 2.72; 95%CI 1.17, 6.31) than high physical activity level, with the lowest nutrition knowledge score (OR 2.66; 95%CI 1.22, 5.77) than higher nutrition knowledge score, and in urban residents (OR 1.90; 95%CI 1.08, 3.36) than rural residents (Table 4). Skipping both of these meals a few times a week, in comparison to never skipping, was significantly associated with low and moderate FAS, low or moderate physical activity level, and a screen time higher than >2 h/day and also moderate/high pHDI (less likely than low pHDI, OR 0.77; 95%CI 0.61, 0.97).

Skippers of both of these meals a few times a week were more likely to be overweight/obese (OR 1.37; 95%CI 1.06, 1.78) in comparison to never skippers (Table 5). For frequent skipping both of these meals and overweight/obesity or central obesity analysis, no significant association was found in the logistic regression.

## 4. Discussion

We estimated that approx. 44% of Polish teenagers usually skipped both breakfast and a meal at school a few times a week while 4.5% frequently skipped both of these meals. Predictors of skipping breakfast and/or any meal at school in Polish teenagers were female gender, age over 12 years, urban residence, lower family affluence, lower nutrition knowledge, higher screen time, and lower physical activity. The association of skipping these meals with diet quality scores was unclear. Skipping both breakfast and any meal while at school a few times a week was associated with increased risk of general adiposity while skipping breakfast frequently was associated with increased risk of general and central adiposity. However, no relation was found between frequent skipping of both of these meals and markers of adiposity or thinness.

Although our sample was not a national representative, it was large and covered the entire territory of Poland, so we can predict the percentage of students consuming or skipping breakfast and/or a meal at school for the population of Polish teenagers aged 11–13 years. We estimated that approx. 52% of students every day consumed both breakfast and any meal at school, approx. 44% skipped both of these meals a few times a week, while 4.5% frequently skipped the first two meals during the day. Considering number of Poles at age 11–13 years [45], it can be estimated that over 490,000 of them may start the school day and learn on an empty stomach. It is hard to compare the study outcomes with others due to a lack of similar studies. When breakfast and a meal at school were considered separately, we estimated that approx. 30% of students skipped at least one day per week either breakfast or a meal at school, including approx. 17% of those frequently skipping breakfast and approx. 13% of those frequently skipping a meal at school. A higher or wider range of the percentage of Polish teenagers (9 to <19 years) skipping breakfast at least one day per week (28%–52%) or any meal at school (near 38%) has been previously reported [32,33,46]. Across Europe, generally more children and adolescents from central or southern than northern countries skipped breakfast at least one day per week, e.g., from Greece (46%–48%), Hungary (38%–48%), and Slovenia (51%–52%) vs. northern countries, e.g., Finland (approx. 20%) [16,47,48].

Our study revealed that the list of predictors of skipping both breakfast and any meal at school was similar to the list of predictors of skipping those meals considered alone, with few exceptions. Urban residence was associated with skipping both of these meals or skipping a meal at school alone but not associated with breakfast skipping, while lower socioeconomic status and female gender were associated with breakfast skipping but not associated with skipping a meal at school alone. In regard to female gender, our findings are consistent with many other studies [10,20,28,30,49]. In Norwegian adolescents, it was found that skipping meals was associated with living in a one-parent family, having parents with low education, and being a male [50].

An association of lower family affluence with skipping the first two meals a day can be explained by opposite social mechanisms. Poland, as a developed country, is free of hunger per se, so lack of food cannot be considered as the reason for skipping meals. However, in Poland there are significant regional and social inequalities which can result in limited access to food at home for teenagers from low-income families [45]. Secondly, the poor economic situation of the family may reduce the care taken to adolescents by caregivers, focusing their activities on earning money [31,32]. On the other hand, children from poorer families are supported in Poland by various social programs, including free school meals [26]. Thus, our findings have highlighted the importance of school nutrition (lunch) programs in equalizing the development chances for children from vulnerable social groups by improving the regularity of meal consumption. The regularity of breakfast consumption is mostly dependent on the family environment, and a lack of daily breakfast consumption by teenagers may be an indicator of a worse family situation.

Three predictors—low-active lifestyle, lower nutrition knowledge, and teenagers’ age over 12 years—can be considered universal because they were strongly related to skipping both meals considered together or alone. In general, our findings are in line with studies previously reported. In adolescents across the world, a significant association has been found between skipping those meals and older adolescents’ age [30,48,49], urban residence [7], lower nutrition knowledge [51,52,53], lower level of education [48,50], lower family affluence [31,54], lower physical activity level [47,48,49], longer time spent sitting and/or longer screen time [21,22], and also having a television in one’s bedroom [21]. Summing up, the current findings indicate that special attention should be paid to promoting shortening screen time and increasing physical activity and teenagers’ nutrition knowledge, which are relatively easily modifiable correlates. It may be speculated that such activities could contribute to more regular consumption breakfast and/or a meal at school, but new intervention studies are needed to find causality.

No clear association was found in regard to diet quality and skipping breakfast and/or a meal at school. A lower chance of a healthier diet was shown for teenagers frequently skipping a meal at school as well as teenagers skipping both of these meals a few times a week. For skipping breakfast a few times a week, two opposite relations were shown—a lower chance of a healthier diet and a lower chance of a less healthy diet. This unclear finding can be explained, first, by the regional specificity of Polish teenagers. Secondly, the use of two simple diet quality scores (each consisted of four food items) which provide a lack of possibility to broadly characterize the diet quality—this may be a limitation of the study and the reason for the inconsistency of our study with other previously reported studies [2,3,4].

The findings provide new insight into the importance of skipping both breakfast and any meal while at school a few times a week. It was shown that skipping both of these meals a few times a week was associated with higher general adiposity in Polish teenagers (overweight/obesity prevalence higher by 37%) and lower thinness prevalence (by 44%). Furthermore, a similar association of frequent skipping breakfast with higher markers of general and central adiposity (by 63%–89%) and lower thinness prevalence (by 48%–55%) was shown. In regard to skipping breakfast, our study is compatible with many others reporting that breakfast skippers are more likely to be centrally obese or overweight (by 60%–90%) than regular breakfast consumers [11,15,20,28,35,51,55]. This may be explained by two mechanisms which have caused an energy imbalance and a tendency for excessive accumulation of fat tissue. Skipping morning meals (one or two) could worsen satiety control and/or promote increased amounts of food consumption and energy intake later in the day [9,17,28]. This supposition is drawn on the basis of outcomes related to breakfast skipping due to a lack of studies related to skipping both breakfast and any meal while at school. A lower inhibition of appetite during the day in breakfast skippers, when compared to regular breakfast consumers, was reported in school-age children and adults [17,56]. Some studies documented that breakfast skippers consumed more caloric meals in the afternoon (by 20%–78% compensation at lunch) than regular breakfast consumers, probably to compensate for a lack of the morning meal, although daily energy intake was not significantly different between subjects skipping breakfast or not [57,58]. In contrast, a few studies showed that adolescents skipping breakfast had an overall lower energy intake than regular breakfast consumers [19,52].

There was no significant association found between the frequent skipping of both of these meals and markers of either general or central adiposity or thinness. This unexpected finding, which suggests that an energy balance is maintained, is difficult to explain—why frequently skipping both of these meals (breakfast: 4–7/week; a meal at school: 3–5/week) was not associated with abnormal body weight while less frequent skipping both of these meals (‘only’ a few times a week) was associated with higher general adiposity and lower thinness prevalence. This could have resulted from two contradictory mechanisms—one, related to the consumption of more caloric meals in the afternoon (as discussed above), and the other, related to lower physical activity to maintain energy balance [28,38,56]. In the present study, teenagers who frequently skipped two morning meals were much less physically active, i.e., were 3–4 times more likely to spend time with low physical activity and/or screen time ≥4 h/day in comparison to others. Further studies are needed to determine whether this unexpected finding, i.e., the lack of association of frequent skipping of both of these meals with abnormal body weight, is nation-specific for Polish teenagers only or if it is a more general rule.

The strength of the study is its relatively large sample (above 1500) which allows a strong statistical analysis to be conducted and specific associations to be found between variables in teenagers within a narrow age range (11–13 years). Since the sample was not randomly selected, but widely reflects the sociodemographic status of Polish society, including teenagers from urban and rural areas and those with lower and higher socioeconomic status there is a good basis for the creation of generalizations. We applied simple obesity measures (BMI, WHtR), widely used in many epidemiological studies, with well-established procedures to collect and interpret data in an international context, despite some interpretative limitations [18,35,44]. In view of this, future studies should consider the use of more advanced methods of adiposity measurement.

The main limitation of the study is the lack of data related to other meals consumed per day and the energy load of those meals and the whole diet. For this reason, fully explaining the association between skipping the first two meals a day, energy balance and body composition is limited. The data are cross-sectional observations thus an association between variables can be found, but no causal effect can be demonstrated. To describe the teenagers’ diets, we used a short form of the FFQ and two simple diet quality scores. There is evidence of many advantages of using brief dietary tools, although several shortcomings have been reported, e.g., the data relies heavily on memory, therefore declining cognitive ability may result in errors when reporting on food frequency consumption; the food list cannot cover all the foods consumed by respondents, which may lead to underreporting; the use of a questionnaire causes some uncertainty due to social desirability bias, especially in females who may provide more socially acceptable answers [59,60]. To describe sedentary and active behaviors, a questionnaire with simple questions was used. This allowed respondents to be ranked into categories of low and high physical activity levels but did not allow precise measuring of physical activity. When time spent in front of a screen was measured, since it was not possible to cover all types of screens used (e.g., the time spent with a smartphone), screen time could be under-reported. Future studies should consider using a long form of the FFQ with a broader list of food items to better describe the usual food consumption and/or applying other methods of lifestyle assessment to fully describe the teenagers’ dietary behaviors and measure physical activity (e.g., use of accelerometry) instead of self-reported data [59,61]. However, to date, there is no validated long form of FFQ, which has been developed for Polish children or adolescents.

## 5. Conclusions

We revealed a large percentage of Polish teenagers (approx. 44%) who usually skipped both breakfast and a meal at school a few times a week. As predictors of skipping these meals, nutrition knowledge-related, socioeconomic and lifestyle correlates were identified. Three predictors (low-active lifestyle, lower nutrition knowledge and teenagers’ age over 12 years) can be considered universal because they were strongly related to skipping both meals, considered together or alone. Therefore, special attention should be paid to promoting shortening screen time and increasing physical activity and teenagers’ nutrition knowledge, which are relatively easily modifiable correlates.

The study shows that skipping both of these meals a few times a week was associated with general adiposity and also strengthens previous evidence showing the association of frequent skipping breakfast with general and central adiposity. Further studies should be focused on explaining the mechanism related to the lack of association of frequently skipping both breakfast and a meal at school with abnormal body weight and compared to other Europeans to also determine whether this unexpected finding is nation-specific for Polish teenagers only or if it is a more general rule.

## Figures and Tables

**Figure 1 nutrients-11-01563-f001:**
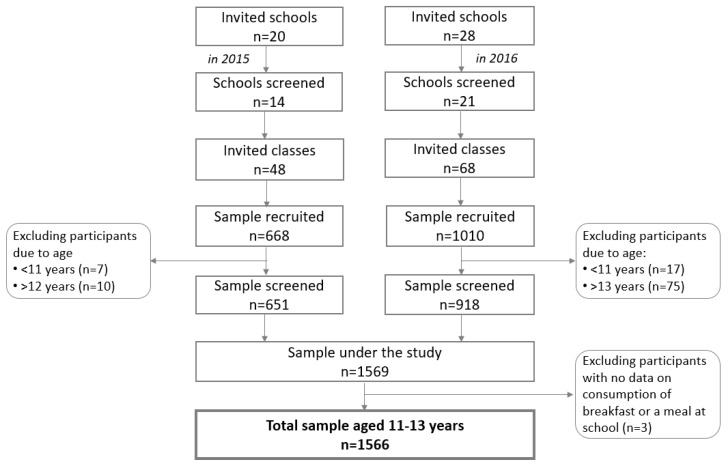
Sample collection.

**Table 1 nutrients-11-01563-t001:** Categorizing skipping breakfast and a meal at school considered together.

Skipping of Breakfast (Number of Days/Week)	Skipping of a Meal at School (Number of School Days/Week)		
	0/week	1–2/week	3–5/week
0/week	Never skipping	Skipping a few times a week	Skipping a few times a week
1–3/week	Skipping a few times a week	Skipping a few times a week	Skipping a few times a week
4–7/week	Skipping a few times a week	Skipping a few times a week	Frequent skipping

**Table 2 nutrients-11-01563-t002:** Categorizing total physical activity based on data regarding physical activity at school and leisure time.

Physical activity at School	Physical Activity at Leisure Time		
	Low	Moderate	Vigorous
Low	Low	Low	Moderate
Moderate	Low	Moderate	Moderate
Vigorous	Moderate	Moderate	High

Notes: Physical activity at school: low (most of the time in a sitting position, in class or on breaks), moderate (half the time in a sitting position and half the time in motion), vigorous (most of the time on the move or on classes related to high physical exertion); physical activity at leisure time: low (more time spent sitting, watching TV, in front of a computer, reading, light housework, a short walk to 2 h a week), moderate (walking, cycling, gymnastics, working at home, or other light physical activity performed 2–3 h/week), vigorous (cycling, running, working at home, or other sports activities requiring physical effort over 3 h/week).

**Table 3 nutrients-11-01563-t003:** Distribution of subjects skipping breakfast and/or a meal at school, socioeconomic correlates, nutrition knowledge, and lifestyle. (Number of subjects and percentages summing up to 100% in rows or columns.)

Characteristics	Total		Skipping Breakfast				Skipping a Meal at School				Skipping Both Breakfast and a Meal at School			
	*n*	%	Never	A Few Times a Week	Frequently	*p*	Never	A Few Times a Week	Frequently	*p*	Never	A Few Times a Week	Frequently	*p*
Sample percentage (%)	1566	100.0	70.0	12.6	17.4		69.0	18.1	12.9		51.9	43.6	4.5	
Gender						ns				***				ns
boys	759	48.5	72.7	11.1	16.2		64.7	20.6	14.7		49.9	45.9	4.2	
girls	807	51.5	67.4	14.0	18.6		72.9	15.8	11.2		53.8	41.5	4.7	
Age (years)						ns				ns				**
11	259	16.5	72.6	12.0	15.4		72.0	16.1	11.9		55.6	40.5	3.9	
12	1153	73.6	70.7	12.4	16.9		69.3	18.2	12.4		52.3	43.8	3.9	
13	154	9.8	60.4	14.9	24.7		61.0	20.8	18.2		42.2	48.1	9.7	
Residence						ns				*				ns
rural	630	40.2	69.4	13.8	16.8		69.2	20.5	10.3		51.2	45.6	3.2	
urban	936	59.8	70.4	11.8	17.8		68.8	16.6	14.6		52.4	42.3	5.3	
Family Affluence Scale						****				*				****
high	401	25.6	78.3	9.5	12.2		73.0	15.6	11.4		60.8	34.8	4.5	
moderate	782	50.0	70.5	13.3	16.2		69.3	19.0	11.8		51.8	44.6	3.6	
low	381	24.4	60.4	14.2	25.5		63.9	19.2	16.8		42.8	50.9	6.3	
Nutrition knowledge score						***				****				**
higher	467	29.8	74.5	10.7	14.8		73.7	18.8	7.5		56.3	41.3	2.4	
moderately-low	615	39.3	70.4	13.8	15.8		68.9	17.4	13.7		53.3	42.3	4.4	
lowest	483	30.9	65.0	12.8	22.2		64.4	18.4	17.2		45.8	47.6	6.6	
Physical activity level						****				****				****
low	153	9.8	60.1	11.8	28.1		58.2	23.5	18.3		40.5	49.7	9.8	
moderate	921	58.9	68.9	14.5	16.5		66.7	19.3	13.9		49.3	46.3	4.3	
high	490	31.3	74.9	9.2	15.9		76.6	14.1	9.4		60.2	36.7	3.1	
Screen time (hours/day)						****				****				****
<2	725	46.4	77.7	9.7	12.7		74.8	15.3	9.9		61.2	35.4	3.3	
2 to <4	538	34.4	65.8	15.4	18.8		68.2	21.0	10.8		47.8	48.5	3.7	
≥4	301	19.2	58.8	14.6	26.6		56.8	19.3	23.9		37.0	54.3	8.7	
pHDI														
low	1079	69.2	66.8	78.7	71.7	***	66.2	73.9	78.7	***	64.9	73.3	78.6	***
moderate	466	29.9	31.9	21.3	27.9		32.4	26.1	21.3		33.4	26.6	21.4	
high	15	1.0	1.3	0.0	0.4		1.4	0.0	0.0		1.7	0.1	0.0	
nHDI						**				**				**
low	1469	93.9	94.2	97.0	90.1		94.1	95.4	90.1		94.5	94.1	84.3	
moderate	91	5.8	5.3	3.0	9.9		5.7	4.6	8.4		5.3	5.4	15.7	
high	5	0.3	0.5	0.0	0.0		0.2	0.0	1.5		0.2	0.4	0.0	

Notes: Sample size may vary in variables due to missing data; categories of FAS: low (0–4 points), moderate (5–6 points), high (7 points); categories of nutrition knowledge score: the lowest (0–4 points), moderately-low (5–7 points), higher (8–18 points); categorizing of physical activity was based on data regarding physical activity at school and leisure time—details are given in Table 2; pHDI: pro-Healthy Diet Index; nHDI: non-Healthy Diet Index; categories of pHDI and nHDI: low (<33.33% points), moderate (33.33 to <66.66% points), high (≥66.66% points); skipping meals: ‘never’—consumption of breakfast 7 days/week, consumption of a meal at school 5 school days/week, ‘a few times a week’—consumption of breakfast 4–6 days/week, consumption of a meal at school 3–4 school days/week, ‘frequently’—consumption of breakfast 0–3 days/week, consumption of a meal at school 0–2 school days/week; statistically significant (chi-square test): * *p* < 0.05; ** *p* < 0.01; *** *p* < 0.001; **** *p* < 0.0001; ns: statistically insignificant.

**Table 4 nutrients-11-01563-t004:** Association of skipping breakfast and/or a meal at school with socioeconomic correlates, nutrition knowledge, and lifestyle in teenagers. (Adjusted odds ratios and 95% confidence intervals; multivariate models.)

Characteristics	Skipping Breakfast (Referent: Never)				Skipping a Meal at School (Referent: Never)				Skipping Both Breakfast and a Meal at School (Referent: Both Never)			
	A Few Times a Week		Frequently		A Few Times a Week		Frequently		A Few Times a Week		Frequently	
	OR	95%CI	OR	95%CI	OR	95%CI	OR	95%CI	OR	95%CI	OR	95%CI
Girls (ref.: boys)	1.52 *	1.10, 2.09	1.44 *	1.09, 1.91	0.67 **	0.51, 0.88	0.78	0.56, 1.07	0.90	0.73, 1.12	1.27	0.75, 2.15
Age (years)												
12 (ref.: 11)	1.04	0.68, 1.59	1.14	0.77, 1.68	1.13	0.78, 1.65	1.06	0.69, 1.65	1.10	0.83, 1.47	1.25	0.59, 2.67
13 (ref.: 11)	1.29	0.68, 2.43	2.04 *	1.17, 3.56	1.28	0.73, 2.25	1.95 *	1.06, 3.59	1.38	0.88, 2.16	5.13 **	1.88, 14.01
Urban residence (ref.: rural)	0.82	0.60, 1.12	1.05	0.79, 1.39	0.80	0.61, 1.04	1.44 *	1.03, 2.01	0.88	0.71, 1.09	1.90 *	1.08, 3.36
Family Affluence Scale												
moderate (ref.: high)	1.64 *	1.09, 2.47	1.54 *	1.06, 2.23	1.25	0.89, 1.75	0.99	0.65, 1.51	1.52 **	1.17, 1.98	0.76	0.40, 1.47
low (ref.: high)	1.90 **	1.19, 3.03	2.45 ****	1.64, 3.66	1.31	0.88, 1.95	1.36	0.88, 2.10	1.90 ****	1.39, 2.60	1.53	0.77, 3.06
Nutrition knowledge score												
moderately-low (ref.: higher)	1.32	0.90, 1.95	1.08	0.76, 1.53	0.96	0.70, 1.34	1.86 **	1.21, 2.86	1.01	0.78, 1.32	1.99	0.93, 4.24
lowest (ref.: higher)	1.36	0.89, 2.08	1.51 *	1.05, 2.17	1.02	0.71, 1.45	2.17 ***	1.39, 3.40	1.26	0.96, 1.67	2.66 *	1.22, 5.77
Physical activity level												
moderate (ref.: high)	1.50 *	1.04, 2.18	0.94	0.69, 1.29	1.65 **	1.20, 2.26	1.53 *	1.05, 2.24	1.46 **	1.15, 1.85	1.28	0.67, 2.44
low (ref.: high)	1.25	0.66, 2.36	1.41	0.87, 2.29	1.93 *	1.17, 3.20	1.96 *	1.11, 3.49	1.53 *	1.01, 2.31	2.72 *	1.17, 6.31
Screen time (hours/day)												
2 to <4 (ref.: <2)	1.83 ***	1.29, 2.59	1.78 ***	1.29, 2.46	1.42 *	1.06, 1.92	1.16	0.79, 1.69	1.70 ****	1.34, 2.14	1.40	0.74, 2.64
≥4 (ref.: <2)	1.94 **	1.26, 2.99	2.82 ****	1.96, 4.06	1.47 *	1.01, 2.14	2.59 ****	1.77, 3.81	2.30 ****	1.71, 3.09	3.93 ****	2.10, 7.37
Moderate/high pHDI (ref.: low)	0.57 **	0.39, 0.83	0.93	0.68, 1.27	0.77	0.57, 1.05	0.69 *	0.47, 1.00	0.77 *	0.61, 0.97	0.71	0.38, 1.34
Moderate/high nHDI (ref.: low)	0.38 *	0.16, 0.90	1.23	0.74, 2.03	0.66	0.35, 1.23	1.09	0.61, 1.93	0.76	0.48, 1.21	1.90	0.87, 4.15

Notes: Sample size may vary in variables due to missing data. Odds ratio adjusted for confounders: gender, age (years), residence (categorical variable), Family Affluence Scale (points), nutrition knowledge (points), physical activity (categorical variable), screen time (categorical variable), excluding the modelled variable from confounders set, respectively; categories of FAS: low (0–4 points), moderate (5–6 points), high (7 points); categories of nutrition knowledge score: the lowest (0–4 points), moderately-low (5–7 points), higher (8–18 points); categorizing physical activity was based on data regarding physical activity at school and leisure time—details are given in Table 2; pHDI: pro-Healthy Diet Index; nHDI: non-Healthy Diet Index; categories of pHDI and nHDI: low (<33.33% points), moderate/high (≥33.33% points); skipping meals: ‘never’—consumption of breakfast 7 days/week, consumption of a meal at school 5 school days/week, ‘a few times a week’—consumption of breakfast 4-6 days/week, consumption of a meal at school 3–4 school days/week, ‘frequently’—consumption of breakfast 0–3 days/week, consumption of a meal at school 0–2 school days/week; statistically significant (Wald’s statistics): * *p* < 0.05; ** *p* < 0.01; *** *p* < 0.001; **** *p* < 0.0001.

**Table 5 nutrients-11-01563-t005:** Distribution of subjects and the association of skipping breakfast and/or a meal at school with adiposity markers in teenagers. (Adjusted odds ratios (ORs) and 95% confidence intervals (95%CI); multivariate models.)

	Sample Percentage (%)						Adjusted Odds Ratio and 95% Confidence Interval					
	BMI-For-Age Categories ^a^				Central Obesity ^b^		BMI-For-Age Categories (Ref.: Normal)				Central Obesity (Ref.: Lack)	
Characteristics	Thinness	Normal Weight	Overweight/Obesity	*p*		*p*	Thinness		Overweight/Obesity			
							OR	95%CI	OR	95%CI	OR	95%CI
Total sample	9.8	65.5	24.7		12.1							
Skipping breakfast				****		**						
never	11.7	67.3	21.1		10.3		1.00		1.00		1.00	
a few times a week	5.4	67.0	27.6		15.7		0.45*	0.23, 0.89	1.25	0.86, 1.81	1.67	1.05, 2.66
frequently	5.3	57.6	37.0		16.8		0.52*	0.29, 0.94	1.89 ***	1.38, 2.58	1.63 *	1.09, 2.44
Skipping a meal at school				*		ns						
never	10.4	67.2	22.3		11.9		1.00		1.00		1.00	
a few times a week	10.0	61.0	29.0		12.7		0.90	0.64, 1.26	1.20	0.87, 1.65	0.85	0.56, 1.30
frequently	5.8	63.7	30.5		13.2		0.93	0.62, 1.37	1.21	0.84, 1.75	0.86	0.53, 1.41
Skipping both breakfast and a meal at school				****		*						
never	12.5	67.9	19.7		10.3		1.00		1.00		1.00	
a few times a week	7.0	63.5	29.6		14.6		0.56**	0.38, 0.83	1.37*	1.06, 1.78	1.24	0.89, 1.74
frequently	6.0	59.7	34.3		10.4		0.52	0.18, 1.51	1.70	0.96, 3.03	0.80	0.34, 1.90

Notes: Sample size may vary in variables due to missing data; skipping meals: ‘never’—consumption of breakfast 7 days/week, consumption of a meal at school 5 school days/week, ‘a few times a week’—consumption of breakfast 4–6 days/week, consumption of a meal at school 3–4 school days/week, ‘frequently’—consumption of breakfast 0–3 days/week, consumption of a meal at school 0–2 school days/week; BMI: body mass index; ^a^ BMI-for-age categorized according to gender-specific BMI cut-offs for teenagers [44]: thinness BMI < 18.5 kg/m^2^; normal weight BMI = 18.5–24.9 kg/m^2^; overweight/obesity BMI ≥ 25 kg/m^2^; ^b^ central obesity identified as waist-to-height ratio ≥0.5 according to Ashwell et al. [18]; odds ratio adjusted for confounders: gender, age (years), residence (categorical variable), Family Affluence Scale (points), nutrition knowledge (points), physical activity (categorical variable), screen time (categorical variable); statistically significant (chi-square test for distribution or Wald’s statistics for odds ratios): * *p* < 0.05; ** *p* < 0.01; *** *p* < 0.001; **** *p* < 0.0001; ns: statistically insignificant.

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
