# Peer review of "Skipping Breakfast and a Meal at School: Its Correlates in Adiposity Context. Report from the ABC of Healthy Eating Study of Polish Teenagers"

_nutrients, 2019, doi:10.3390/nu11071563_

Reviewer 1 Report

Thank you for giving me the opportunity to read and review this interesting manuscript about the frequency of skipping breakfast and/or school meal and its association with various factors. The study has many strengths, for example the large sample size. The authors have also included several background factors in their models as potential confounders/covariates. However, there are also shortcomings that the authors need to carefully reflect. I think the main problem with the manuscript is that is not clear all the time if the authors want to predict overweight using meal skipping and other factors or do they want to predict meal skipping using weight status and other factors. Even though this is a cross-sectional study, I think it is conceptually important to be consistent with the whole idea of the paper. Because of the cross-sectional data, the authors should also be very careful not to make any causal inferences. There were also some grammatical grievances, and I think the manuscript would significantly benefit from language revision.

Abstract

Point 1: It would be nice to explain shortly how each of the predictors in the models were measured (self-report, weighted, etc.). You state that respondents reported usual consumption of breakfast etc., but they also reported other variables, right?

Point 2: Lines 39–41: You suggest that similar activities can be taken to improve the regularity of the consumption of these meals due to the similarity of predictors. Those predictors were female gender, age over 12 years, urban residence, lower family affluence, lower nutrition knowledge, higher screen time and lower physical activity. In my opinion, there are two problems with this statement. First of all, the predictors were associated with the frequency of consuming these meals, that is, there is an association, but based on this study, you cannot make assumptions regarding the direction of the association nor the causality. Second, not all these “predictors” can be changed: age and gender we cannot change, and I would find it implausible also to think researchers/public health advocates could change urban residence or family affluence. I think you could focus on the “predictors” that could plausibly be changed, that is, screen time and physical activity (whether these are the reason for skipping breakfast or not, I’m not sure).

Point 3: Lines 41–43: This study does not show that skipping these meals increase general adiposity! This study only shows an association between skipping meals and adiposity. There might, for example, be reverse causality: the overweight/obese participants might skip meals in order to lose weight. They might also underreport their food consumption.

Introduction

Point 4: Lines 50–53: It could be beneficial if you could add the age ranges for the participants in different studies, since school-age covers a broad age range. In addition, including the country where the study was conducted could be nice, because culture can affect breakfast and its consumption frequency. Lines 54–63: I don’t think it is important to discuss the potential mechanisms here in detail, since it is not the main aim of this particular study. Maybe these lines could be moved to discussion? It might be a more appropriate context for this kind of speculation.

Point 5: Line 83: Should the reference here be #15?

Materials and Methods

Point 6: Line 104: Please state clearly that the academic centres were in Poland. You could also state if they covered the whole of Poland or just specific parts of Poland.

Point 7: Lines 107–108: What does “All data were collected at schools instead of regular school lessons” mean? Does this mean the data were collected during school lessons? Who collected the data, teachers or researchers/research assistants? Did they receive any training?

Point 8: Did you apply for an ethical statement/permission? From where?

Point 9: How many schools were contacted and how many of them agreed to participate? Were all the classes with 11–13-year-old pupils from the participating schools invited or just part of the classes? How did you choose the invited schools and classes if not randomly?

Point 10: Line 125: Are you describing exclusion criteria or inclusion criteria?

Point 11: I think chapter 2.3 Breakfast and a meal at school could be reorganized. Now it’s rather challenging to follow. First you speak of frequency of meal consumption but you don’t state the exact question that the participants answered or the response options they had. The second paragraph contains this information, and I think it could be wise to start with this. On lines 134–145 you have a lot of detailed information, but it is unclear, whether these were instructions given to the participants or whether the researchers used these guidelines to clear the data.

Point 12: Line 152: There seems to be a typo, the second category should be 1–2/week?

Point 13: Table 1: Does this table offer any information that is not stated in the text? If not, it could be omitted.

Point 14: When explaining the FFQ, it could be stated right away if the FFQ was validated or not. I would consider moving the FFQ-related validation results (lines 215–230) to the section in which you describe the FFQ. However, it must be noted that reproducibility (test-retest) is only a part of the validity of a certain measure (it can be stated that the participants seem to answer similarly at two separate time points but it does not prove that the FFQ is able to measure food consumption truthfully).

Point 15: Please state clearly in the text which categories were combined (for example line 186).

Point 16: Why do you use the term sedentariness if you measured screen time? Would it make more sense to use the term screen time? I would consider using the term screen time throughout the text. Also, why did you only ask how much time did the participants spent watching TV or in front of a computer? Don’t you think they used smartphones or tablets? Please state also the response options in the text.

Point 17: I think it would be nice to report the questions/statements used to measure nutrition knowledge for example in a supplementary table. You could also report the number/proportion of participants who answered correctly into each of the questions.

Point 18: For me, Figure 2 is not very useful. I don’t know if I am interpreting it correctly. The Figure, in my mind, suggests that you are predicting obesity with the other variables. However, your tables suggest that you are predicting skipping breakfast using the other variables (including obesity). Thus, the purpose of this figure is a bit unclear to me.

If I look at the aims of the study, it seems that you are interested in 1) the association of several background factors (nutrition knowledge, lifestyle and socio-economic status) and skipping breakfast/school meal and 2) to assess the association of skipping meals with obesity. Thus, I think you should have a model that answers to question 1 and then maybe another models answering the second question. Anyway, I think the manuscript would improve significantly, if the conceptual model behind your research aims was more clearly stated and followed in the text.

Results

Point 19: Table 2: Please indicate the statistical methods used to derive the p values for each of the variables (in the footnote, for example). You should also include cut-off points for each category (for example low, moderate and high) for each of the variables. The readers should be able to interpret the table on itself.

Point 20: Table 3: Please state if this is a multivariable model or not (were all the variables entered into the same model or were their associations investigated separately).

Point 21: I would consider using ORs in the text (instead of stating that teenagers with a screen time of 4 or more hours/day were 159% more likely to frequently skipping a meal.

Discussion

Point 22: You should discuss more the limitations introduced by the methods. For example the FFQ has several shortcomings, and the questionnaire used to assess physical activity seems quite light. Also your screen time measure is not able to capture all types of screens used.

Point 23: Line 342–344: “Skipping a few times a week of both breakfast and any meal while at school increased the risk of general adiposity while frequent skipping breakfast increased the risk of general and central adiposity.” I am confused as to whether you used skipping meals as a predictor of obesity or the other way around. I am aware that the study is cross-sectional and thus, it’s kind of the same thing, but conceptually I think it is important that the aim of the study, the methods used to investigate it as well as the conclusions are in line with each other. I understood that you were interested in finding out if skipping breakfast predicted obesity (the aim of the study) but from Table 3 I think you predicted the odds of skipping breakfast. This must be clear throughout the manuscript.

Point 24: Lines 392–394: Your findings do not indicate that increasing physical activity, reducing sedentary time (screen time!) and increasing nutrition knowledge would lead to more regular consumption of breakfast/school meal. However, I do think you can discuss these as potential points of intervention. Then you could maybe speculate on how for example increasing physical activity could contribute to more regular breakfast consumption. Maybe you could find intervention studies showing an increase in breakfast consumption?

Point 25: Lines 405–407: This was not shown! It was shown that obesity/overweight was associated with skipping breakfast, but which came first, is impossible to say from this cross-sectional data.

Point 26: Lines 423–425: Again, this conclusion cannot be drawn from this study. At least you should mention that there is a possibility for reverse causality.

Conclusions

Point 27: Lines 466–468. This study did not show that skipping both meals increases general adiposity. There was an association, which could be to that direction, but it could also be the other way around. Please be careful when reporting the results from cross-sectional data.

Point 28: Lines 468–469: I don’t think it is safe to say, based on this study, that you could predict obesity by identifying teenagers who skip meals. In addition, since your study categorized overweight and obese participants into the same category, I think throughout the text, you should speak about overweight or obese participants, not obese participants.

Author Response

Dear Reviewer,

We are very grateful that we have the opportunity to resubmit the revised version of our manuscript (ID: nutrients-535562) entitled ‘Skipping breakfast and a meal at school: its correlates in adiposity context. Report from the ABC of Healthy Eating study of Polish teenagers’.

We greatly appreciate the time and efforts taken by the Reviewers and the Editor to review our manuscript. We have addressed all issues indicated in the review reports, and believe that the revised version can meet the journal publication requirements.

We would like to underline that issues regarding study sample and methods were previously published by Hamulka et al. The effect of an education program on nutrition knowledge, attitudes toward nutrition, diet quality, lifestyle, and body composition in Polish teenagers. The ABC of Healthy Eating project: Design, protocol, and methodology. Nutrients, 2018, 10, 1439, doi:10.3390/nu10101439, although we followed all Reviewer’s suggestions and added more details.

Please find our responses to the Reviewer’s comments attached. The manuscript has been corrected for language errors, using professional editing (native speaker) and proof-reading service. All changes in the manuscript are highlighted in blue font.

Yours Sincerely,

Jadwiga Hamulka and Natalia Wojtas (Ulewicz)

Reviewer 2 Report

The authors present an interesting study whose aims were to identify nutrition knowledge-related, lifestyle (including diet quality, physical activity and sedentariness) and socioeconomic correlates of skipping breakfast and a meal at school, considered together or alone, and to assess the association of skipping these meals with obesity markers in Polish teenagers.

The manuscript is properly structured and drafted.

The introduction section adequately contextualizes the study and its importance.

The Methods section adequately describes the participant selection process, inclusion criteria, and the final sample size. However, the authors should improve the following aspects:

-        The selection of the sample was conducted in elementary schools (not randomly selected). How schools were selected?  What criteria were considered to choose a school or another?

-          An participant inclusion criteria was “fourth and fifth grade classes of elementary school” Why do authors choose these courses? Authors must include this information in the text.

-          Authors must include more information on all anthropometric measurements indicated, How were these determinations performed? Did the authors follow a standard anthropometric protocol? Were the ISAK guidelines followed? They must specify the protocol used...

The Results section clearly describes the main findings of the study. The figures and tables are pertinent and improve the compression of the contents.

The Discussion section is consistent with the results. The authors discuss adequately the results obtained, comparing with studies of similar characteristics.

The bibliography consulted is pertinent and current.

Author Response

Dear Reviewer,

We are very grateful that we have the opportunity to resubmit the revised version of our manuscript (ID: nutrients-535562) entitled ‘Skipping breakfast and a meal at school: its correlates in adiposity context. Report from the ABC of Healthy Eating study of Polish teenagers’.

We greatly appreciate the time and efforts taken by the Reviewers and the Editor to review our manuscript. We have addressed all issues indicated in the review reports, and believe that the revised version can meet the journal publication requirements.

We would like to underline that issues regarding study sample and methods were previously published by Hamulka et al. The effect of an education program on nutrition knowledge, attitudes toward nutrition, diet quality, lifestyle, and body composition in Polish teenagers. The ABC of Healthy Eating project: Design, protocol, and methodology. Nutrients, 2018, 10, 1439, doi:10.3390/nu10101439, although we followed all Reviewer’s suggestions and added more details.

Please find our responses to the Reviewer’s comments attached. The manuscript has been corrected for language errors, using professional editing (native speaker) and proof-reading service. All changes in the manuscript are highlighted in blue font.

Yours Sincerely,

Jadwiga Hamulka and Natalia Wojtas (Ulewicz)

Round  2

Reviewer 1 Report

Thank you for revising the manuscript. I think the quality of the paper has improved significantly. I still have some minor remarks.

Abstract

Have you considered using ORs in the abstract also?

Introduction

Lines 57–60: ”… higher consumption of energy-dense snack foods, which was caused by lower intakes of vitamins, minerals, fibre and protein.” Is this correctly said?

Materials & Methods

Since your sample is recruited as classes and schools, have you considered using multilevel models? That would take into account the fact that students from one class might resemble each other more than students from different classes/schools.

Results

I like that you have used 95% CIs in the Tables, I think it gives more information than p values. I would have used them also in the text, I think the p values are not that important.

Table 3 & 4: Do you need the abbreviation ns (statistically insignificant) in the Tables? I don’t think that is needed, so that could maybe be omitted from the footnote.

Author Response

Dear Ms.  Vikki Li, Dear Reviewer,

We are very grateful that we have the opportunity to resubmit the revised version of our manuscript (ID: nutrients-535562) entitled ‘Skipping breakfast and a meal at school: its correlates in adiposity context. Report from the ABC of Healthy Eating study of Polish teenagers’.

We greatly appreciate the time and efforts taken by the Reviewer #1 and the Editor to review our manuscript. We have addressed all issues indicated in the review reports, and believe that the revised version can meet the journal publication requirements.

Please find our responses to the Reviewer comments attached. All changes in the manuscript are highlighted by using the "Track Changes" function.

Yours Sincerely,

Jadwiga Hamulka and Natalia Wojtas (Ulewicz)
